

# Genetic diversity and organic waste degrading capacity of *Hermetia illucens* from the evergreen forest of the Equatorial Choco lowland

María Fernanda Pazmiño[1,2], Ana G. Del Hierro[2] and Francisco Javier Flores[1,3]

[1] Departamento de Ciencias de la Vida y de la Agricultura, Facultad de Ingeniería en Biotecnología, Universidad de las Fuerzas Armadas-ESPE, Quito, Pichincha, Ecuador
[2] Laboratorio de Investigación Aplicada—Biotecnología, Instituto Nacional de Biodiversidad-INABIO, Quito, Pichincha, Ecuador
[3] Centro de Investigación de Alimentos, Facultad de Ciencias de la Ingeniería e Industrias, Universidad Tecnológica Equinoccial, Quito, Pichincha, Ecuador

## ABSTRACT

Globally, microplastics (MP) represent a growing burden for ecosystems due to their increasing presence at different trophic levels. In Ecuador, the lack of waste segregation has increased the quantity of waste, primarily organics and plastics, overloading landfills and water sources. Over time, plastics reduce in size and silently enter the food chain of animals, such as insects. The black soldier fly (BSF) larvae, *Hermetia illucens* (Linnaeus, 1758), is a species with devouring behavior used for waste management because of its beneficial qualities such as fly pest control, biomass production, and rapid organic waste degradation. Studies have uncovered the insect's ability to tolerate MP, and consider the possibility that they may be able to degrade polymers. For the first time in Ecuador, the present study characterized *H. illucens* using the sequences of different molecular markers. Finally, *H. illucens*' degrading capacity was evaluated in the presence of MP and decaying food residues, resembling landfill conditions.

## INTRODUCTION

In the last century, synthetic plastics have become widely available and essential to human lifestyles (*Dris, Agarwal & Laforsch, 2020*; *Evode et al., 2021*). However, the plastic production rate of 400 Mt year$^{-1}$ is higher than its degradation time (*Chamas et al., 2020*). Plastic bags take approximately 20 years to degrade, while packaging and bottles can take up to 500 years, making them almost indestructible because of their complex decomposition (*World Wildlife Fund, 2018*). Only 9% of plastics are recycled, 12% are incinerated, and 79% remain at disposal sites or are dumped in the environment (*Geyer, Jambeck & Law, 2017*).

Over time, plastics fractionate into tiny particles of less than five milimeters (mm), called microplastics (MP) (*Chia et al., 2021*). It has been discovered that aquatic and terrestrial organisms easily ingest particles of this ubiquitous material, threatening ecosystems because

Corresponding author
Ana G. Del Hierro,
anagdelhierro@gmail.com

of their increasing presence at different trophic levels (*Cho et al., 2020*; *Wong et al., 2020*; *Baho, Bundschuh & Futter, 2021*; *Wang et al., 2021*). The most common plastics found in the market include polyethylene (PE), polypropylene (PP), polyvinyl chloride (PVC), and polystyrene (PS) (*Bond et al., 2018*). Polystyrene, is a material used for packaging, has a linear polymeric carbon structure with phenyl groups that are very stable and require a great deal of energy to break and decompose (*Bond et al., 2018*). Polyethylene is the material from which single-use bags are made. It also has a linear polymeric carbon structure (*Evode et al., 2021*). Polylactic acid (PLA) is used to manufacture biodegradable plastic bags. This material is derived from corn starch and has a polymeric structure of lactic acid monomers, a molecule used by living organisms (*Zaaba & Jaafar, 2020*).

Waste overproduction and the lack of technology investment, especially in lesser developed and developing countries, cause the loss of essential services and benefits provided by nature, such as soil for food harvesting, water, and clean air, to mention a few (*Mmereki, Baldwin & Li, 2016*). Ecuador, a middle-income South American country, produced 5.4 million metric tons of waste in 2017 (*Ministerio del Ambiente, Agua y Transición Ecológica, 2012*). Despite being recognized as the first country to establish nature's rights in its constitution in 2008 (*Tanasescu, 2013*), it has not fully implemented waste separation strategies to protect the environment. Approximately 60% of the daily waste produced in Ecuador is organic waste and 20% is potentially recyclable inorganic solid waste, including plastics (*Ministerio del Ambiente, Agua y Transición Ecológica, 2012*). Thus, waste management technology is urgently needed to valorize waste and divert it from landfill.

Insect bioconversion is a cost-effective and environmentally friendly method for transforming waste into biomolecules with high biological and economic value (*Shelomi, 2020*; *Da Silva & Hesselberg, 2020*; *Skrivervik, 2020*; *Franco et al., 2022*). The black soldier fly (BSF), *Hermetia illucens*, is one of the most studied insect species because of its outstanding ability to decompose fruits, vegetables, food scraps, vegetation, and even agricultural waste such as animal manure (*Barragan-Fonseca, Dicke & Van Loon, 2017*; *Liu, Wang & Yao, 2019*; *Surendra et al., 2020*). In this regard, insect farms are biorefinery factories in which every part of the process is recycled (*Azagoh, Hubert & Mezdour, 2015*). *H. illucens'* larvae are effective waste recyclers that consume a wide variety of organic wastes (*Nana et al., 2019*). Depending on the type of food used as a substrate, its life cycle lasts for approximately 30 days. *H. illucens* evolves from the egg to five larval instars, transforming into pupae and becoming a mature fly (*Lalander et al., 2019*). Throughout this time, the larvae grow and consume all the organic matter possible, storing in their body macro and micronutrients (*Barragan-Fonseca, Dicke & Van Loon, 2017*; *Lalander et al., 2019*; *Surendra et al., 2020*; *Franco et al., 2022*).

Full-grown larvae are composed of 30%–52% dry matter (*Surendra et al., 2020*) and approximately 39% dry weight fat (*Nana et al., 2019*). At the fifth instar of development, larvae are collected and ready for use as feed for animals (*Dortmans et al., 2017*). If processed, the larvae are dried and ground into a high-quality, sustainable meal to replace conventional protein sources for food and animal feed, most commonly soybean meal and fish meal (*Giannetto et al., 2020*; *Hartinger et al., 2022*). Another application is to extract

fat from larvae and apply this subproduct as soy oil replacement for animal supplements (*Kim et al., 2022*), personal care products such as shampoo, detergent, and soap (*Franco et al., 2022*), and biodiesel production (*Surendra et al., 2016*; *Lee, Yun & Goo, 2021*). In addition, insect's frass stores valuable nutrients to serve as soil fertilizer (*Liu, Wang & Yao, 2019*; *Lopes, Yong & Lalander, 2022*). *H. illucens* last instar larvae (pupae) are a source of chitin and chitosan that are suitable for industrial and biomedical applications (*Triunfo et al., 2022*). On the other hand, recent studies have highlighted *H. illucens* larvae's ability to remediate polluted biomass (*Bulak et al., 2018*), cadmium (*Zhang et al., 2021*) and mercury (*Attiogbe, Ayim & Martey, 2019*).

Insects have adapted to unfavorable conditions and may have developed gut microbiota to metabolize some types of components, such as heavy metals and even plastics. *Cho et al. (2020)* reared *H. illucens* on PS and PE- MP substrates. This study found that the survival capacity of *H. illucens* was not affected, whereas both microplastic substrates reduced pupation and substrate rates. *Brandon et al. (2018)* reported that mealworms degrade PE and mixtures (PE + PS). *Beale et al. (2022)* concluded that *Tenebrio mollitor, H. illucens* and *Galleria mellonela* exposed to different plastic (PET, PE, PS, expanded PE, PP, and PLA) expressed different plastic degradation pathways.

*H. illucens*, which is likely native to tropical regions (*Kaya et al., 2021*), has not been documented in Ecuador, and molecular analysis of the population has not been conducted. To identify Ecuadorian *H. illucens* and compare them to *H. illucens* from other locations, we analyzed three barcodes (COI, ITS2 and 28S rDNA) from individuals captured in Puerto Quito and Nanegalito. Our first hypothesis was that the Ecuadorian *H. illucens* population belongs to *H. illucens* and its closest relatives are *H. illucens* from neighboring countries (Peru and Colombia). Furthermore, considering the current Ecuadorian waste segregation practices, where plastics and organic materials end up together in landfills, our second hypothesis was that MP ingestion affects the development of *H. illucens* and its ability to biotransform waste. The objective of this study was to molecularly characterize Ecuadorian *H. illucens* and assess its capacity to biotransform waste residues containing MP.

## MATERIALS & METHODS

### Sample collection

The samples were collected as part of the Framework Agreement on Access to Biological and Genetic Resources of the Scientific Investigation granted by The Ministry of Environment, Water and Ecological Transition-MAATE (Permit number MAATE-DBI-CM-2021-0173). *H. illucens* adults with different body coloration phenotypes were collected in 2021 from Puerto Quito (coordinates 0°08′04.1″N 79°16′32.0″W) and Nanegalito (coordinates: 0°06′05.1″N 78°43′39.8″W), Ecuador (Fig. 1). Adult flies were trapped using nets and collected in mesh insect cages. *H. illucens* females were captured in the wild while attempting to lay eggs near the pig manure. Eggs were obtained from adult females of Puerto Quito. Female flies were gently pressed between the folds of a small piece of cardboard to induce oviposition until all the eggs were laid. Subsequently, a small piece of cardboard with approx. 500–900 eggs per fly was placed on the banana puree until arrival at the laboratory

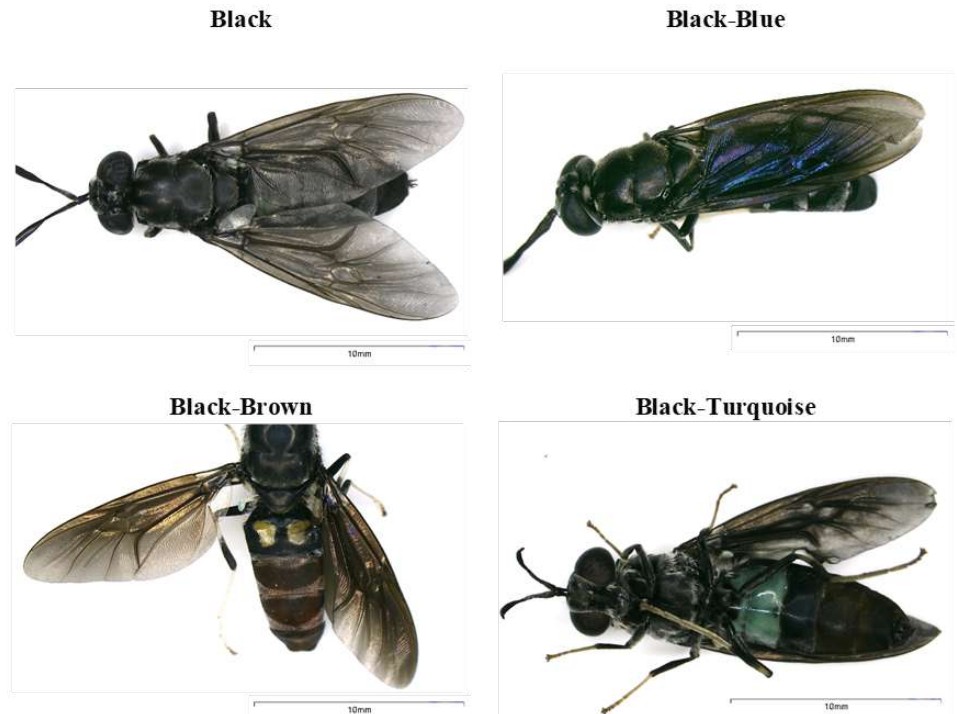

**Figure 1** Body coloration phenotypes identified in black soldier flies found in Ecuador.

in a growth chamber for larval emergence (Fig. 2). During transportation to the laboratory, adults were preserved in 96% alcohol at 4 °C. They were then stored at −20 °C prior to DNA extraction for molecular characterization.

## Molecular characterization

Genomic DNA was extracted from individuals using the PureLink Genomic DNA Mini Kit (Invitrogen, Waltham, MA, USA), following the manufacturer's instructions. The quality and concentration of the resulting DNA were evaluated by absorbance measurements using a Nanodrop 2000/2000c spectrophotometer (Thermo Fisher Scientific, Waltham, MA, USA). The extracted DNA was stored at −20 °C until further use.

A partial sequence of the cytochrome c oxidase I (COI) gene, the internal transcribed spacer 2 (ITS2), and a partial sequence of the structural ribosomal RNA for the large subunit (28S rDNA) gene were amplified from the total DNA of each individual by polymerase chain reaction amplification (PCR) (*Ståhls et al., 2020*). PCR conditions were obtained from previous research, taking into account the conditions of the commercial Taq polymerase used: DreamTaq Green PCR Master Mix (2×; Thermo Scientific, Waltham, MA, USA). For COI, the PCR conditions were pre-denaturing for 180 s at 95 °C, and 35 cycles of 30 s at 95 °C, 45 s at 52 °C, and 60 s at 72 °C, and a final extension for 300 s at 72 °C (primers: LepF1 5 forward: 5′ ATTCAACCAATCATAAAGATATTGG 3′; LepR1 5 reverse: 5′ TAAACTTCTGGATGTCCAAAAAATCA 3′) (*Khamis et al., 2020*). For ITS2, PCR conditions were pre-denaturing for 180 s at 95 °C, followed by 35 cycles of 30 s at 95 °C, 30

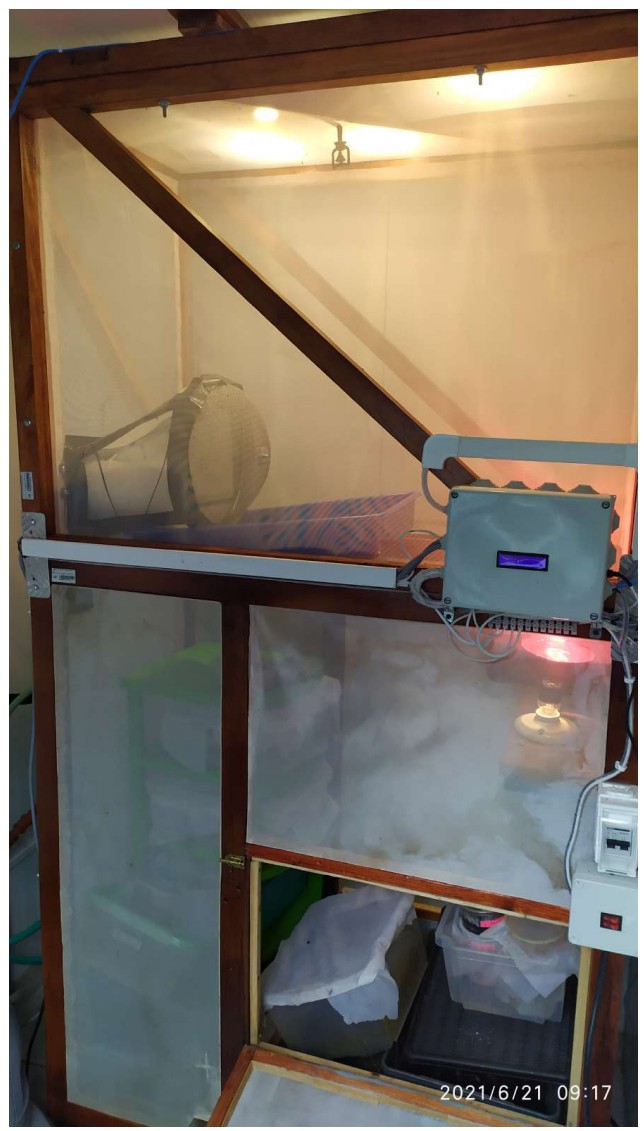

**Figure 2** *H. illucens* **growth chamber.** A temperature and humidity-controlled chamber was constructed to emulate the natural conditions of the experiment. The cage was designed for rearing *H. illucens* eggs, larvae and adults. It was supported by a wooden structure lined with tulle-type mesh fabric walls. To control humidity, a water sprinkler system was turned on periodically. For temperature control, an infrared heating bulb was programmed to turn on when a detector noticed a drop in temperature (below 15 °C). Inside the cage, larvae were reared in plastic containers covered with tulle fabric.

s at 50 °C, and 60 s at 72 °C, and a final extension for 300 s at 72 °C (primers: ITS2a forward: 5′ TGTGAACTGCAGGACACAT 3′; ITS2b reverse: 5′ TATGCTTAAATTCAGGGGGT 3′) (*Ståhls et al., 2020*). For 28S rDNA, the PCR conditions were pre-denaturing for 180 s at 95 °C, followed by 35 cycles of 30 s at 95 °C, 30 s at 41 °C, 60 s at 72 °C, and a final extension for 300 s at 72 °C (primers: 28S_P03_F forward: 5′ TTYRGGAYACCTTYDGGAC 3′; 28S_P03_R reverse: 5′ GGTTTCCCCTGACTTCDACCTGATCA 3′) (*Ståhls et al., 2020*).

The total volume for all PCR reactions was 25 µL with 12.5 µL of master mix, 0.4 µM of each primer, and 100–200 ng of genomic DNA. The fragments were observed on a 1.5% agarose gel (Invitrogen, Waltham, MA, USA), with 1X TAE buffer (Invitrogen) and 4 uL of the intercalating agent SYBR Safe (Invitrogen) and compared with the 100 bp DNA ladder TrackIt (Invitrogen). Amplicons were sent to Macrogen™ (Seoul, South Korea) for bidirectional Sanger sequencing.

The resultant sequences were assembled using Geneious Prime® v2021.2.2 (https://www.geneious.com) and compared with the National Center for Biotechnology Information (NCBI) GenBank database to identify close homologs using the Basic Local Alignment Search Tool (BLAST). A partial COI gene phylogenetic tree was constructed using the Bayesian method to analyze the relationship of the Ecuadorian population with *H. illucens* individuals from other countries. Ecuadorian individual sequences were aligned with Genbank-NCBI references using Geneious Prime v2021.2.2. BEAST software v1.10.4 (*Suchard et al., 2018*) was used to construct the Bayesian tree. The outgroups used to construct the phylogenetic tree were *H. coarctata* (MT434000) and *H inflata* (MG967891). The phylogenetic tree image was obtained using FigTree v.1.4.4 (*Suchard et al., 2018*) and edited with Microsoft PowerPoint.

An uncorrected distance (*p*-distance) analysis of the partial COI gene sequences was performed using MEGA11: Molecular Evolutionary Genetics Analysis version 11 (*Tamura, Stecher & Kumar, 2021*) to calculate genetic differences between *H. illucens* clades that were identified in the Bayesian phylogenetic tree.

### *H. illucens* rearing

The eggs collected were reared in a controlled growth chamber at 23.8 ± 2.2 °C, 12 h:12 h L:D photoperiod and relative humidity of 41.3 ± 12.6% (Fig. 2). After two days of laying, the substrate (banana puree) was renewed to avoid contamination with fungi or mites. Eggs hatched after 5–6 days of laying, and then six-day-old larvae were starved for twenty-four hours to stimulate feeding in the presence of MP to perform a waste degradation assay (*Brandon et al., 2018*; *Cho et al., 2020*).

### Waste degradation assay

A total of 360 larvae were transferred to 12 polypropylene containers (12 cm d × 6 cm h) to hold 30 larvae per container. Each container was covered with a tulle-like fabric mesh using rubber bands to prevent larvae from escaping and other insects and contamination from entering.

The experiment was performed in triplicate using four different diets: a banana puree control and three mixtures containing 95% banana puree and 5% MP (PS polystyrene, PE polyethylene oxo-biodegradable bags, or PLA corn starch bags) (Supplementary Material I). In the first week, 30 g of the substrate was fed to 7-day-old larvae. Every seven days, 30 g of freshly prepared substrate was administered. An additional 20 mL of water was added every three days to prevent the substrate from drying out (*Cho et al., 2020*).

Plastics used in this experiment (PS, PE, and PLA) were reduced to MP by manual fragmentation (grating and cutting), obtaining sizes smaller than 5 mm:2.12 ± 1.01 mm

for PLA, 1.71 ± 1.40 mm for PE and 1.06 ± 0.76 mm for PS, measured with Fiji software (*Schindelin et al., 2012*) (Supplementary Material II).

Ten larvae from each of the 12 containers were weighed twice a week to calculate the biomass (*Cho et al., 2020*; *Raimondi et al., 2020*). Before weighing, the larvae were randomly sampled by hand, cleaned with distilled water and dried with paper towels. The experiment was concluded when the larval pupation ratio reached 25% (*Bruno et al., 2019*). In addition, larvae were separated once a week to collect insect frass and stored at −80 °C (*Brandon et al., 2018*; *Yang et al., 2018*). After acknowledging that the biomass data followed a normal distribution, summary measures were obtained for the larval biomass weight. An analysis of variance, ANOVA test, was also performed, along with a Tukey test to see if there were significant differences between treatments.

## RESULTS

### Molecular characterization

COI, ITS2, and 28S rDNA partial sequences, of about 657, 453, and 270 bp in length, respectively, were obtained for nine *H. illucens* individuals from Ecuador (Puerto Quito and Nanegalito), the Neotropical biogeographic area (Table 1). Identity percentages for these sequences ranged from 97.72% to 100% based on NCBI-BLAST alignment analysis.

Because the ITS2 and 28S rDNA sequences showed intraspecific invariance between isolates, phylogenetic analysis was only performed for the COI partial gene sequences (Fig. 3). Aggrupation formed by this analysis indicated that *H. illucens* from Ecuador clumped together in one individual clade. Except for isolate G, which associated with *H. illucens* from Venezuela and Mexico. According to the Bayesian analysis, we divided *H. illucens* sequences into three groups according to the clades shown in Fig. 3.

Clade 1 included the isolates: ON783031 (Ecuador), ON783032 (Ecuador), ON783033 (Ecuador), ON783034 (Ecuador), ON783035 (Ecuador), ON783036 (Ecuador), ON783038 (Ecuador), ON783039 (Ecuador), MT520663 (Thailand), LR778159 (Bhutan), MT178509 (Singapore), KM928149 (Canada), LR792233 (Panama), MT483918 (Costa Rica), LR778209 (Venezuela), LR778211 (Venezuela), ON783037 (Ecuador), LR778208 (Mexico), MT178512 (Poland) and MT178496 (Italy).

Clade 2: LR792234 (French Guiana), FJ794358 (South Korea), JN308284 (Papua New Guinea), LR792223 (Malaysia), LR792255 (Australia), LR778204 (Colombia), LR792230 (Peru), LR792238 (Bolivia), LR778205 (Brazil), LR792235 (Brazil), LR778158 (Ghana), LR792240 (Paraguay), LR778203 (Bolivia), LR792231 (Peru) and LR792239 (Paraguay).

Clade 3: MT178493 (France), LR792260 (Spain), MN868766 (Portugal), MT483914 (China), MT178503 (Vietnam), MT178497 (Indonesia), KY817115 (Russia), LR792263 (South Africa), LR792261 (Switzerland), MT520686 (USA), LR792259 (Kenya) and MG682545 (India).

The *p*-distance results showed a distance of 0.0422 between clades two and three and 0.0419 between clades one and three, while clades one and two were the closest, with a distance of 0.0254 (Table 2). On the other hand, distance analysis within clades ranged from 0 in clade 3 to 0.02 in clade 1, showing high identity for intraclade COI nucleotide sequences (Table 3).

**Table 1   NCBI Accession numbers for the *Hermetia illucens* partial COI, ITS2 and partial 28S rDNA gene sequences obtained.**

| Individual | Collection location | NCBI accession number | | |
|---|---|---|---|---|
| | | COI | ITS | 28S |
| A | Puerto Quito | ON783031 | ON783702 | ON782650 |
| B | Puerto Quito | ON783032 | ON783703 | ON782651 |
| C | Puerto Quito | ON783033 | ON783704 | ON782653 |
| D | Nanegalito | ON783034 | ON783705 | ON782654 |
| E | Puerto Quito | ON783035 | ON783706 | ON782652 |
| F | Puerto Quito | ON783036 | ON783707 | ON782655 |
| G | Nanegalito | ON783037 | ON783708 | ON782657 |
| H | Nanegalito | ON783038 | ON783709 | ON782656 |
| I | Puerto Quito | ON783039 | ON783710 | ON782658 |

**Table 2   Genetic distance (*p*-distance) of *Hermetia illucens* partial COI sequences between clades stated with Bayesian phylogenetic analysis.**

| | 1 | 2 | 3 |
|---|---|---|---|
| CLADE 1 | | 0.0045 | 0.0075 |
| CLADE 2 | 0.0254 | | 0.0069 |
| CLADE 3 | 0.0419 | 0.0422 | |

**Notes.**
Standard error estimate(s) are shown and were obtained by a bootstrap procedure (500 replicates) which involved 47 nucleotide sequences. All ambiguous positions were removed for each sequence pair (pairwise deletion option). There were a total of 687 positions in the final dataset.

## Waste degradation assay

At the end of the experiment, the mean weight of 10 larvae (30 days) was 0.278 g for the control, 0.190 g for 5% PE, 0.264 g for 5% PS, and 0.392 g for 5% PLA. No prepupal formation was observed and the average larval biomass did not reach the expected value in any of the treatments, including the control. According to the Tukey's test, there were significant differences between the larval biomass reared with PLA starch bags and PE bags, but there were no significant differences between the other assays (Table 4, Fig. 4).

## DISCUSSION

For the first time in Ecuador, DNA information from *H. illucens* was obtained. Genetic variability among individuals isolated from different locations was analyzed using COI, ITS2, and 28S rDNA molecular barcodes. Our results are consistent with those of a previous study by *Ståhls et al. (2020)* that identified a high level of intraspecific genetic diversity (4.9%) for the COI gene among isolates sequenced from various parts of the world, with 56 different haplotypes. In contrast, ITS2 and 28S rDNA partial gene sequences showed intraspecific invariance. Genetic variability of the COI gene between populations supports its use as a molecular marker for characterizing *H. illucens* (*Park et al., 2017*; *Ståhls et al., 2020*).

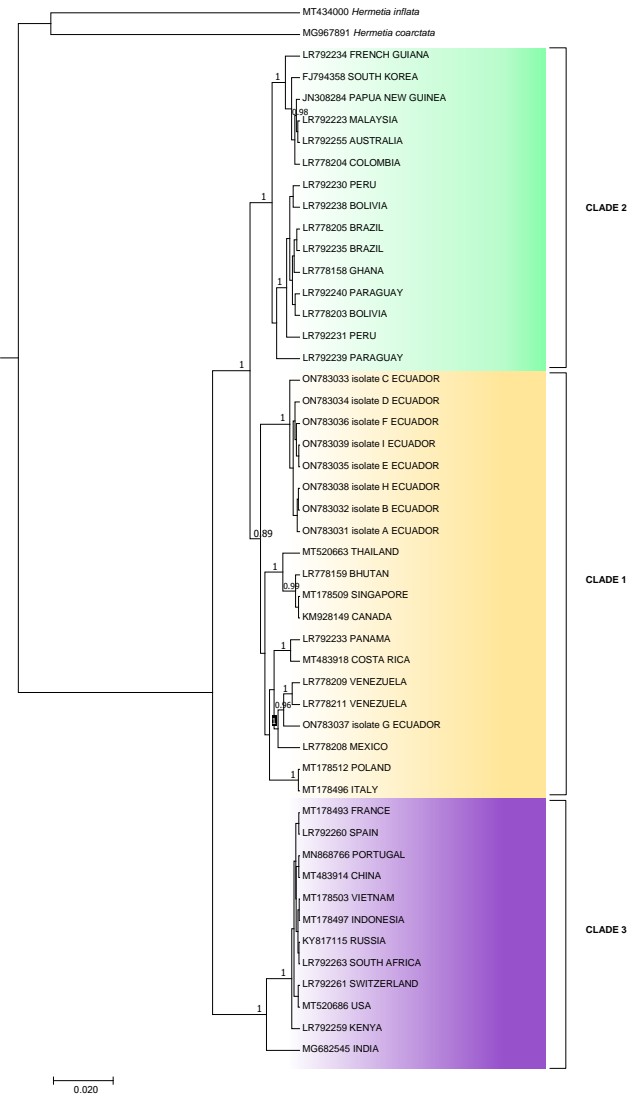

**Figure 3  Bayesian molecular phylogenetic tree for *Hermetia illucens* COI partial gene sequences.**
Posterior support values probabilities >0.85 (located on the nodes) were calculated with BEAST software:
GTR substitution model, strict molecular clock, MCMC chain length 1,000,000, sampling rate 1,000,
burn-in 10%. The bottom left scale indicates the calculated genetic distance.

The individuals from Puerto Quito and Nanegalito, Ecuador, showed no phylogenetic
spatial structures. However, isolate G clumps closer to isolates from Venezuela and Mexico
than to the Ecuadorian population. *P*-distance results indicated the most significant
genetic divergence between clade 1, which included the *H. illucens* Ecuadorian population,
and clade 3, and between clade 2, which also comprises specimens from Latin America,
compared to clade 3 (Fig. 3).

Clade 1 was represented by isolates from Ecuador and clade 3 represented isolates from
China, Vietnam, Indonesia, India, the USA, Russia, Europe, and Africa. Isolates from

**Table 3** Genetic distance (*p*-distance) of *Hermetia illucens* partial COI sequences within clades stated with Bayesian phylogenetic analysis.

|  | d | S.E. |
| --- | --- | --- |
| **CLADE 1** | 0.02 | 0.00 |
| **CLADE 2** | 0.01 | 0.00 |
| **CLADE 3** | 0.00 | 0.00 |

**Notes.**
Standard error estimate (S.E.) was calculated by a bootstrap procedure (500 replicates) which involved 47 nucleotide sequences. All ambiguous positions were removed for each sequence pair (pairwise deletion). There were a total of 687 positions in the final dataset.

**Table 4** Analysis of variance for larvae biomass.

**Analysis of variance**

| Variable | $N$ | $R^2$ | Ad $R^2$ | Coefficient of variation |
| --- | --- | --- | --- | --- |
| Biomass | 12 | 0.61 | 0.46 | 25.13 |

**Analysis of variance table (SS type III)**

| Source of Variation | Sum of squares | Degrees of freedom | Mean Squares | $F$ | *p*-value |
| --- | --- | --- | --- | --- | --- |
| Model | 62666.25 | 3 | 20888.75 | 4.19 | 0.0468 |
| Treatment | 62666.25 | 3 | 20888.75 | 4.19 | 0.0468 |
| Error | 39912.67 | 8 | 4989.08 | | |
| Total | 102578.92 | 11 | | | |

**Notes.**
The critical $F$ value (3.86) is lower than the test $F$ obtained, therefore biomass means had great differences between treatments.
$N$, sample size; $R^2$, percentage of variation in the response; Ad $R^2$, Adjusted $R^2$; SS, sum of squares.
Significance level 0.05.

Ecuador may have a more remarkable similarity with *H. illucens* from Thailand, Singapore, Bhutan, Europe and Latin America (Costa Rica, Venezuela, Mexico) that constitute clade 2. In addition, *Khamis et al. (2020)*, *H. illucens* phylogenetic analysis from West Africa (Nigeria and Ghana) formed a distinct group; samples from Thailand and the United States were closely related and samples from Uganda were grouped separately. All the samples from Australia, the Netherlands, South Africa, Kenya, the United States, and China were grouped together. In our study, clade 2 of the *H. illucens* COI partial sequences from Australia, South Korea, Malaysia, Ghana and South America (Colombia, Peru, Bolivia, Paraguay, Brazil, and French Guiana) clustered together.

Similarly, *Park et al. (2017)* found that high molecular differentiation between populations indicated a limit on the dispersal of *H. illucens* in a regional manner that could be affected by climatic conditions. The native range of *H. illucens* is thought to be in Central America and the northern regions of South America, suggesting a limited spread of the Ecuadorian population in Latin America. The apparent spread of this species through South Asian coastlines and islands could have been accidental introductions led by goods maritime transport (*Marshall, Woodley & Hauser, 2015*; *Guilliet et al., 2022*).

*Guilliet et al. (2022)* performed complete COI gene analyses. The most remarkable diversity of *H. illucens* was found in Latin America, supporting the hypothesis of geographic origin. Indeed, the estimated divergence times between the different haplotypes, were considered low compared to the calculated amount of diversification. Compared to
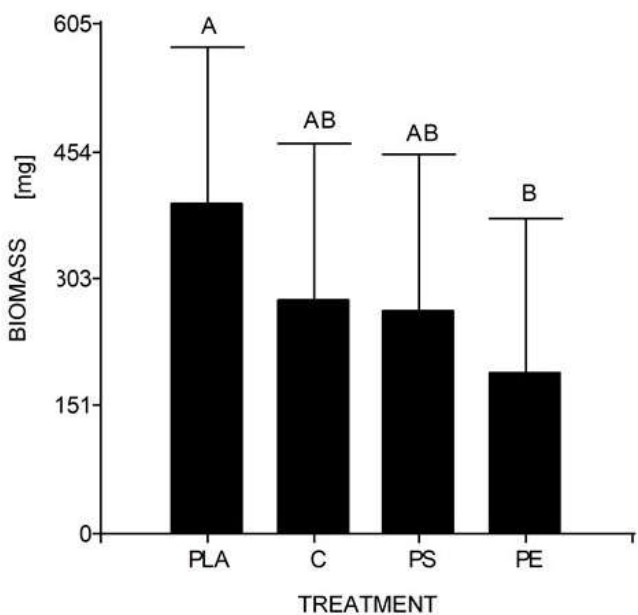

**Figure 4  Mean comparison graph between the biomass values obtained within the different degradation test treatments.** The intervals represent the minimum significant difference (MSD) value obtained in the Tukey test. Treatments PS and C had no significant differences between each other. Nevertheless, the PLA microplastic treatment significantly produced the highest insect biomass, and the PE treatment significantly manifested the lowest insect biomass.

other species, these events make it impossible to assume that such divergence is due to a single introduction of the species in South America. In addition, they reported 30 haplotypes among 55 COI sequences sampled in South America, opening the possibility of additional unknown and novel sequences in the region. The Ecuadorian *H. illucens* could be an example of a distinct haplotype, showing differences among other Latin American individuals of the same species. A haplotype study is recommended to determine whether the Ecuadorian *H. illucens* represents a new haplotype that could support the hypothesis of this insect origin in South America.

Regarding the second study's hypothesis on the development of *H. illucens* and its ability to biotransform waste mixed with MP, we found that the values of larval biomass and pupal ratio do not resemble those of other studies. The lowest weight/biomass was observed in *H. illucens* larvae reared on food waste treated with 5% PS and PE MP (Fig. 4). Furthermore, 5% PLA-reared larvae produced the highest amount of biomass. Due to fermentation promoted by the bacteria, the PLA polymer can be transformed into lactic acid and other monomers that serve as food for the larvae. Hydrolyzed PLA products represent an additional easy-to-assimilate food source that could have led to a significant increase in the biomass of the 5% PLA MP treatment. After exposure to moisture, PLA undergoes lysis of its ester groups in its main chain, causing its molecular weight to decrease and releasing soluble oligomers and monomers of lactic acid and glucose (*Zaaba & Jaafar, 2020*).

Additionally, the 5% PE- and PS MP-reared larvae showed no significant differences from the control group. This does not mean that there is no MP impact on larvae, as the biomass of larvae reared at 5% PE was the lowest. Indeed, *Cho et al. (2020)* revealed that larvae fed 400–500 μm powder forms of PS and low-density PE had a low substrate reduction rate (~40%) and a higher pupation ratio (>20%) compared to the control. As contaminants can interfere with hormone production and act as endocrine disruptors, it is unclear whether PE or PS MP affect *H. illucens* larval physiology (*Cho et al., 2020*).

*Cho et al. (2020)* and *Romano & Fischer (2021)* studied the ability of *H. illucens* to tolerate MP. They found that polypropylene MP significantly reduced the pupation ratio (at 65.2% compared to the control at 83.8%) and increased the amount of short-chain fatty acids in larvae (propionic and butyric acid at 0.17% and 0.19%, respectively). A possible change in the gut microbiota may be the answer to these changes. However, no study has proven the ability of *H. illucens* larvae to degrade polymers. An analysis underpinned by *Beale et al. (2022)* exposed *H. illucens* to different plastic diets. The study concluded that the presence of plastic might have caused oxidative stress affecting a wide variety of cellular responses related to the immune response. Finally, they suggested studies based on insect microbiota to specify the effects of plastics on *H. illucens*. Thus, it should be noted that microbial life, which is also present in all ecosystems and significantly in the animal tract, is considered as the key player in the degradation of all types of materials and in the development of insects.

The study of the effect of plastics on the diet and development of commercial insects has caught the attention of the scientific community in recent years. *Mitra & Das (2022)* exposed ten *H. illucens larvae* to three of the same plastic substrates as in the current study (PE, PS and PLA). Nevertheless, their research focused on studying the biochemical impact of plastics using three insect models. In the case of *H illucens*, this study found that the presence of plastic appeared to cause oxidative stress. In addition, the authors highlighted that all plastics caused a downregulation of Vitamin B6 metabolism, suggesting a breakdown in the gut symbiont.

Moreover, *Lievens et al. (2022)* studied the survival effects and bioconversion rate of *H. illucens* when fed food waste streams containing micro- (<5 mm), meso- (5–25 mm), and macroplastics (>25 mm) present in food packaging material (polyvinyl chloride (PVC)). In contrast to our results, using a different type of plastic, their study showed a complete larval cycle and relatively larger larvae than the control. According to these findings, the plastic mixture could have decreased substrate density, facilitated oxygen uptake, and thus positively influenced larval growth.

When proposing the experiment, we expected to have low larval growth, but complete the entire cycle. *Scala et al. (2020)* emphasized that *H. illucens* larvae can adapt growing on more deficient protein diets; however, they produce smaller larvae than richer protein feeds. *H. illucens* is a resilient organism and therefore have the ability to extend its life cycle under unfavorable conditions. Studies such as *Cho et al. (2020)* showed a 24-day cycle of larvae reared in 5% PS and 5% PE with a larval weight between 0.20–0.25 g and a pupation ratio of about 20%.

Regarding the control substrate for our experiment, we chose banana as the control diet because of its sugar, carbohydrate, and fiber content, different from the most common type

of diet used in other studies, the Gainesville diet (30% alfalfa, 50% wheat bran, 20% corn meal). According to *Oonincx et al. (2015)*, in a diet high in carbohydrates, such as bananas, but low in protein and lipid contents, the *H. illucens* cycle took 37 days. For this reason, a banana control was chosen as a comparable control for plastics, expecting a longer lifecycle owing to the lack of nutrients.

The MP substrate may have altered *H. illucens* larvae, which did not seem to be eating feed because of their growth and weakness. When no larval weight changes were observed, the experiment was interrupted at four weeks of larval development, and larvae, substrate and frass samples were collected for further processing. Because the larval instar is a crucial stage for nutrient uptake, the reserves of fat and lean proteins must be sufficient for pupation and adulthood (*Dortmans et al., 2017*). Low-quality substrates have been reported to affect prepupation development (*Lalander et al., 2019*). When feeding a diet different from high-quality organic food, the protein- to-carbohydrate ratio is involved (*Gold et al., 2020*). Feeding larvae with optimal food quality composed of a 21% protein to 21% carbohydrate ratio will ensure growth and cycle development (*Cammack & Tomberlin, 2017*; *Dortmans et al., 2017*). Several studies have highlighted that substrate composition alters the *H. illucens* lifecycle, survival, nutritional composition, and feed conversion (*Barragan-Fonseca, Dicke & Van Loon, 2018*; *Surendra et al., 2020*; *Laganaro, Bahrndorff & Eriksen, 2021*; *Eggink et al., 2022*). Thus, the MP-waste substrate did not provide an adequate diet to complete the insect cycle and development.

After the waste degradation test, a chemical analysis was conducted to determine the bioaccumulation or degradation of plastic in the larval substrate or intestinal tract. Fourier-transform infrared spectroscopy (FTIR) analysis of frass, feed, and larvae was planned to determine wheter there were changes in the functional groups of the polymers to demonstrate possible biodegradation. In addition to the larvae, rearing substrates and their frass were collected. Frass collection was one of the most challenging tasks in this experiment because these particles adhered to the larvae's bodies. *Beale et al. (2022)* found the same limitation in insect frass collection in the jar, suggesting an additional analysis to assess micro and nanoplastics. In addition, screening for organic plastic residues in the insect gut and expelled frass is recommended. A multi-omics analysis of future generations will also demonstrate the actual impact of plastics on the bioaccumulation of this material.

To determine wheter there were chemical changes in the polymer structure of the plastics that might suggest degradation, these components were extracted from the samples for FTIR analysis. Tetrahydrofuran (THF) solvent was used for the PE samples, and dichloromethane (DCM) was used for the PS and PLA samples as described by *Brandon et al. (2018)* and *Yang et al. (2018)*. However, these solvents were too volatile and a small amount (two mL) was used, which made it challenging to obtain extracts for analysis. Hence, we suggest standardizing the extraction of PE and PLA plastic bags and PS containers using the correct amount of solvents. In 2021, the current investigation was limited by COVID-19 pandemic restrictions. Consequently, chemical analyses of the samples could not be completed. We recommend further analysis of MP bioaccumulation and substrate-to-biomass conversion in future studies.

Our experiment included 30 larvae per container to run a reproducible experiment; so that one person could manipulate each larva, quantify them, and weigh them separately for further processing. *Beale et al. (2022)* used ten larvae per jar in a similar experiment. The low number of samples was justified by compliance with the Metabolomics Standard Initiative guidelines, which set a minimum number of biological replicates ($n = 3$) per treatment group. On the other hand, *Cho et al. (2020)* used 100 larvae per container. Based on *Shishkov et al. (2019)*, *H. illucens* exhibits a collective behavior that could help larvae resist limiting situations. They proposed a mathematical model to understand the feeding mechanics of *H. illucens* larvae, where they feed faster in groups, thereby increasing their feeding rate. We believe that more than 30 larvae per container should be used for the MP-waste degradation assay to obtain proper larval weight results; *Beale et al. (2022)* recommended increasing the insect biomass and plastic ratio to obtain better results.

We argue that limited larval growth is due to deficient nutrient intake when feeding on the MP-waste substrate, which delays pupal formation and may be related to larval density. *Barragan-Fonseca, Dicke & Van Loon (2018)* stated that larval density affects developmental time, survival, and larval weight. In fact, they found that larval density and nutrient content in the diet affected larval weight, body composition, and larval performance of *H. illucens*. These results were obtained by feeding 0.6 g of food (dry matter basis) per larva once per week. The larvae were fed chicken feed diluted with cellulose and data were collected until the formation of the first prepupa. Lower densities and higher nutrient contents accelerated larval development. The shortest development period obtained was 13 days, when larval density was 50 larvae per container or 0.31 larvae cm$^{-2}$ and nutrient content was the highest tested (14% protein, 1.8% fat, 46% non-cellulose carbohydrate). The most extended development period was 45 days and it was reported when larval density was high (200–400 larvae per container or 1.23–2.47 larvae cm$^{-2}$, respectively) and nutrient content was low (3.5% Protein, 0.7% Fat, 12% Non-cellulose carbohydrate). Individual larval weights and larval yield per container increased with higher nutrient content and high larval densities (100–200 larvae per container or 0.62–1.23 larvae cm$^{-2}$, respectively).

During the design of a MP-waste degradation experiment it is essential to consider larval density, feed nutritional composition and rearing conditions to obtain meaningful data. Larval density of at least 100–200 larvae per experimental unit should be considered. Banana feed or other more nutritious feed can be used to simulate landfill conditions. Thirty larvae should be randomly sampled every week to record their weight. Once the first pupa appears, the experiment should be terminated to start the analysis phase, where larvae, substrate and frass are sampled for bioaccumulation and conversion chemical tests. The experiment must be performed under controlled temperature and humidity conditions in a rearing chamber. A prolonged larval stage period should be expected (>30 days).

*H. illucens* is one of the most studied insects because of its capacity to convert a wide range of organic wastes. Nevertheless, numerous parameters affect growth performance, complicating comparisons with other studies (*Lievens et al., 2022*). In the case of plastic conversion, information is still limited and further studies are required to answer multiple unanswered questions regarding plastic degradation mechanisms and bioaccumulation (*Li et al., 2021*; *Beale et al., 2022*; *Mitra & Das, 2022*). If human behavior continues to force

insects to be exposed to plastics, food chain changes will affect all living organisms, not only insects.

## CONCLUSIONS

*H. illucens* COI, ITS2, and 28S rDNA barcode sequences were obtained from nine wild individuals collected from Puerto Quito and Nanegalito, Ecuador. A similarity and phylogenetic analysis of these sequences confirmed that these individuals belonged to *H. illucens*. *P*-distance analysis showed that Ecuadorian *H. illucens* tended to group in one clade but may be more related to *H. illucens* in Latin America (Venezuela, Mexico, Costa Rica) and South Asia (Thailand, Singapore, and Bhutan). Haplotype studies on this insect are recommended to corroborate the Latin American origin hypothesis.

The MP degradation test showed no significant differences between the treated and control samples. However, there was a substantial difference in the effect of larval biomass reared with MP between PE bags and corn starch—PLA bags, where the biomass obtained with 5% PE MP was negatively affected. Our results showed that MP could affect the development of *H. illucens* larvae. Thus, this information supports the development of environmental strategies using *H. illucens* for waste biodegradation.

Our study tested the plastic degrading capacity of *H. illucens* to demonstrate the detrimental impact of non-segregating plastics on insects' lives, joining the scientific community's call to find sustainable ways to address the environmental degradation caused by the overproduction of plastics.

## ACKNOWLEDGEMENTS

The authors thank Instituto Nacional de Biodiversidad INABIO, IDgen for providing a Molecular Identification Laboratory for the study, and Universidad de las Fuerzas Armadas—ESPE for all the support.

### Funding

The publication of this article was financed through public research funds of the Ecuadorian Government, INEDITA Research Call. Agreement signed on December 10, 2018, between the Instituto Nacional de Biodiversidad (INABIO) and the Secretaría de Educación Superior, Ciencia, Tecnología e Innovación (SENESCYT) for the execution of the project "Bioconversion of organic waste and plastic from invertebrates of Ecuador", code number 00110378. The funders had no role in study design, data collection and analysis, decision to publish, or preparation of the manuscript.

### Grant Disclosures

The following grant information was disclosed by the authors:
INEDITA Research Call.

Instituto Nacional de Biodiversidad (INABIO) and the Secretaría de Educación Superior, Ciencia, Tecnología e Innovación (SENESCYT).

"Bioconversion of organic waste and plastic from invertebrates of Ecuador": 00110378.

## Competing Interests

The authors declare there are no competing interests.

## Author Contributions

- María Fernanda Pazmiño conceived and designed the experiments, performed the experiments, analyzed the data, prepared figures and/or tables, authored or reviewed drafts of the article, and approved the final draft.
- Ana G. Del Hierro conceived and designed the experiments, performed the experiments, prepared figures and/or tables, authored or reviewed drafts of the article, and approved the final draft.
- Francisco Javier Flores conceived and designed the experiments, analyzed the data, prepared figures and/or tables, authored or reviewed drafts of the article, and approved the final draft.

## Field Study Permissions

The following information was supplied relating to field study approvals (*i.e.*, approving body and any reference numbers):

Access approval to the Genetic Resources of the Scientific Research Project Entitled Scientific Research Project Entitled: "Bioconversion of Organic and Plastic Waste from Invertebrates of Ecuador" between The Ministry of Environment, water and the Ministry of Environment, water and Ecological Transition, was given through The Undersecretariat of Natural the Undersecretariat of Natural Patrimony; and, the National Institute of Biodiversity (MAATE-DBI-CM-2021-0173).

## Data Availability

*Hermetia illucens* sequences (including partial COI, ITS2 and partial 28S rDNA gene sequences) are available at GenBank: ON783031–ON783039, ON783702–ON783710, ON782650–ON782658.

## Supplemental Information

Supplemental information for this article can be found online at http://dx.doi.org/10.7717/peerj.14798#supplemental-information.

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
