# Peer review of "Genetic diversity and organic waste degrading capacity of Hermetia illucens from the evergreen forest of the Equatorial Choco lowland"

_PeerJ, doi:10.7717/peerj.14798_

## Round 0.1 · original submission · Major Revisions

The contribution has been sufficiently reviewed by the exerts in the field, and their comments are attached. After going through the reviewers comments and original MS, it has inferred that the methodology section is the most weaker portion of the MS which thoroughly needs to be strengthen by adding technical details to insure the fundamental concept of repeatability. I do have serous concerns of sampling size and would definitely like to know who the author address this concern. Results may be describe in bit more detail and interpretation of recorded findings must be enriched. Conclusions should be aligned with the findings.

Reviewer 1 ·

Basic reporting

No comment.

Experimental design

My main concern for this study is limitation of number of specimens (n=9) used for DNA analysis. Although author examine 3 genes but those from nuclear (ITS2 and 28S rDNA) show no variation. Therefore, I recommend author to increase number of specimens use for COI sequence analysis.

Validity of the findings

No comment.

Additional comments

This manuscript examines genetic diversity of black soldier fly, Hermetia illucens, using the COI, ITS2 and 28S rDNA sequences. The manuscript also testing the waste degradation efficiency of the larvae of this insect. My main concern for this study is limitation of number of specimens (n=9) used for DNA analysis. Although author examine 3 genes but those from nuclear (ITS2 and 28S rDNA) show no variation. Therefore, I recommend author to increase number of specimens use for COI sequence analysis. Following are my additional comments and suggestions:
1. Title, I suggest to modify to “Genetic diversity and organic waste degrading capacity of Hermetia illucens from evergreen forest of the Equatorial Choco lowland”.
2. Line 47; author species name is needed for first mention.
3. Line 118 – 129; I recommend author to state specifically about the hypothesis or the objective of this study for molecular examination. For example, do the authors would like to compare COI sequences of H. illucens from Ecuador with other part of the world that have already reported? Why additional genes (i.e. ITS2 and 28S rDNA) were used?
4. Lines 125 – 129; different phenotypes of H. illucens are mentioned here but there is no specific hypothesis or objective of this study whether author would like to examine genetic differentiation between different phenotypes?
5. Lines 190 – 193; Arlequin is used for AMOVA analysis? If so, please state this clearly. However, the purpose of AMOVA is to examine genetic differentiation between “groups of population”. According to the results present in line 241 – 244, it seems that there is only the FST between group of sequences identified by phylogenetic analysis. Therefore, I recommend to use K2P or uncorrected p-distance between these groups of sequences that is more appropriate than AMOVA.
6. Lines 226 – 229; Please provide ‘intraspecific genetic divergence” for H. illucens examine in this study (population from Ecuador) and for overall variation based on the whole set of data. Also, report the level of genetic divergence between phylogenetic clades (see above comment). Also, genetic differentiation between morphological forms of H. illucens.
7. Lines 241 – 244; This section can be removed and was replace by level of genetic divergence between phylogenetic clades.
8. Line 255 – 280; This discussion section should be rewrite with the information getting from the new analysis (i.e. intraspecific genetic divergence, genetic differentiation between phylogenetic clades).
9. Line 286 – 290; this is the result, thus should be moved to the Results.
10. Table 2, this table can be removed following above comment (no. 4).
11. Figure 2, Please state clearly which method of phylogenetic tree inference is use for this figure. It is not possible that this figure is represent both maximum likelihood (ML) and Bayesian trees (BA). It must be ML or BA. Both methods possibly produce identical or similar tree topologies but only ML or BA was selected for presentation. If author would like to present both trees they must be in different figures.
12. Figure 2 caption; RaxML is standalone software for ML tree inference. So, it is not true that RaxML was used to calculate bootstrap support by MEGA 7.
13. Figure 3, because only three pairs of the comparisons, it is not necessary to provide this figure.

Reviewer 2 ·

Basic reporting

Basic reporting
Clear and unambiguous, professional English used throughout:
1. Avoid starting a sentence with an abbreviation (line 58, line 137;check text throughout) and be sure to clarify what acronyms represent (line 63; what is an “AMOVA”?).
Literature references, sufficient field background/context provided:
1. Line 110- Remove “Since BSF lack functional mouthparts”. First, as larvae they have functional mouthparts (just saying BSF is not specific enough) and have teeth-like structures on their mouthparts can suck up liquified food (Bruno et al. 2019). Having functional mouthparts and the ability to sting or bite are to different things. This can be removed because it is misleading and the sentence works without it.
Literature references, sufficient field background/context provided.
1. Line 112 needs citations.
2. I think more information about BSF is needed in the introduction–largely because interest in the species is growing–there are more papers that can be included to expand the last sentence. What are the wide applications in the fields listed on line 116 specifically? Are there other environmental benefits (reduction in volatiles and greenhouse gas or obnoxious odors, less resources used.)? What about the alternative–the impacts of inadequate waste disposal on the environment? Can they be used as a feed ingredient-if so, to which species? This is a good place to educate the readers (consider an audience that knows nothing about BSF).

Experimental design

Experiment design
Methods described with sufficient detail & information to replicate.
1. More details are needed about how the BSF were reared prior to the experiment. The methodology for the experiment has a lot of detail, but there are details missing leading up to the experiment. Please include information such as:
a. What containers and cages were used to house the adults, eggs, and larvae? How were eggs collected? What type of ovipositional site was used?
b. How much food was provided to the larvae? What was the larval density? How often were they fed?
c. It is not clear what the degradation (line 65) test is¬–please explain.
d. Please add in any detail that may be missing and consider the perspective of a reader who knows nothing about BSF and how to rear them. How would you tell them to set up their colony before the experiment?
2. Line 148: what are “DNAweres”?
3. Sometimes in the text “BSF” is used, others “H. illucens” is used–although they are referring to the same species, select one and replace the other for consistency purposes.
4. Waste degradation assay: How were the larvae counted, by hand? Gravimetrically? Were the larvae weighed prior to placement on the treatments? How much diet was offered at the beginning of the experiment? More details are needed about how the larvae were set up for the experiment–see suggestions above regard how the larvae were treated prior to the experiment.
5. Line 211- should be: “Ten larvae were weighed twice a week for a biomass calculation”
6. Why 25% prepupation (line 214)?
7. Line 218 “was conducted to determine if significant differences existed between treatments” instead of “was run to know…”
8. Line 242- beginning with “To determine…” check the tense (past vs present) on this sentence– “is” and “was” are used in the same sentence. Also, the next sentence is also in present tense. Change to past tense for all sentences and check manuscript for consistency. Please have the manuscript looked over for grammatical issues before resubmission.
9. Line 248– missing “the” in front of “control. See comment above about having the manuscript edited/reviewed for grammatical errors.

Validity of the findings

Validity of the findings
Conclusions are well stated, linked to original research question & limited to supporting results.
1. Line 268: what is considered “very high?” General terms like high, low (any type of description) are meaningless without some specificity. For this reason, include the value
2. The discussion could be developed more.
a. Further explanation is needed for the paragraph involving phylogenetic spatial structure. Other studies on closely related species would be beneficial to discuss–are these finding common in other flies? What are the implications of the current findings? How can this information be applied to an audience that is not familiar with phylogenetics?
b. What is pupation ratio? Is there a formula? Is it the same as the percentage of individuals that pupated? If so, may be useful to change to “percent pupation” as this is more common terminology in the literature.
c. When discussing larval biomass and pupation, more literature should be cited to defend the argument that larval density played a role on the growth and development of BSF. See Banks et al. 2014- look for other small-scale studies. Comparing the current results to one stud is not sufficient to back up the statements.
d. Same comment as above for the argument about the quality of the resource provided.
e. Line 302- why was the research interrupted? Is there a reason why they were not allowed to feed for longer than 4 weeks? Are there other BSF studies that have observed a longer larval development time?
f. Many BSF studies have a Gainesville diet or chicken feed control. It may be useful to include this as a limitation of the study.
g. Line 298- the feeding mechanisms may be part of why they feed more efficiently in groups but feeding faster does not mean they will acquire adequate nutrition to make it to the next life stage. It would be good to develop part of your discussion about the nutritional part–even if you only compare your control results to other studies. I know there are a few papers about feeding BSF bananas (some with the banana peel; not sure if that was part of the diet used in this study so please clarify that in the methodology).
h. Line 324- is 20% pupation “high”? Seems a very low.
i. Line 325- please explain what an endocrine disruptor is for an audience that knows nothing about insects.
j. Some grammatical issues found (does not include all):
i. Line 326 “larval” instead of “larvae”
ii. Line 328- “conducted” instead of “expected” and “to determine” instead of “to look”
iii. Line 330 “performed” instead of “planned”
iv. Line 332 “were collected”
v. Lines 338, 339. 344-345
k. Line 352- This line makes it seem like BSF cannot consume material with microplastics, but that wasn’t Cho’s findings-they suggest that BSF can be used to manage wastes with microplastics, although treatment efficiency may be lowered. Because of the conflicting findings, line 352 in its current state does not seem accurate if under different circumstances (e.g., different larval density as discussed in the discussion).

·

Basic reporting

This article is well written, well structured and clear to follow. A good list of references has been cited, and sufficient background information has been presented. Raw data (in this case NCBI numbers leading to gene sequences) have been provided.

Experimental design

The article is within the aims and scope of this journal. The study aimed to use molecular barcodes or markers to characterise BSF from Equador and assess their capacity to biotransform waste supplemented with various microplastics. This study's aim is well-defined and meaningful, and the methods used to answer the research objective are well described to allow for replication.

It would have been good if more analysis had been done on the samples obtained from the waste degradation assay. For example, the authors comment that other studies, such as Cho et al., 2020 used 100 larvae whereas 360 larvae were used for the entire experiment in this study. The authors of the current work could have reared more BSF larvae to increase the numbers used in this experiment. Moreover, is there a reason why standard proximate analysis (especially moisture, proteins, and fat contents) were not conducted on the BSF larvae fed with MPs? This information could have added another perspective and insight into the biomass/ weight gained by the larvae. The authors mentioned challenges due to COVID which prevented them from analysing MP bioaccumulation in the larvae. This is understandable. But because some of these plastics (e.g. PLA) are degraded into monomers of glucose and lactic acid, it is not clear if the extraction with organic solvents will necessarily show anything.

Validity of the findings

The title of the manuscript does not reflect the fact that waste degradation of the larvae was studied. Could atuhotrs modify the title to reflect the degradation study objective?

In the introduction, the ability of BSF to degrade plastics is assumed, but no reference to scientific publications is given. Could authors provide proof of BSF's ability to degrade plastics? Studies such as https://www.sciencedirect.com/science/article/pii/S0048969722019337?via%3Dihub come to mind.

In L308-309, the authors mentioned that height and pressure might have affected larvae growth. Can they provide a scientific basis for this statement? In other words, are there any published studies that show that height and pressure affect insect larvae growth?

In L300-304, the authors mentioned that the waste degradation experiment was interrupted at 4 weeks of larval development because pupation (typically taking 15 days) did not occur in these samples. Do authors expect the results to be different if the larvae had fed well and followed the usual growth cycle for BSF?

Based on the above comment, it will be good for the authors to propose how the waste-MP degradation experiment could be done differently to give meaningful and comprehensive data. The authors mentioned this in Lines 399-342 but did not explain what they would do differently.

Additional comments

L105. Please write MP in full in this line. i.e. Microplastic (MP)

L115-116 It is not clear why waste degradation by BSF is linked to applications in medicine. This seems far-fetched.

L115-116 talks about how BSF converts organic wastes into proteins and fat. L118-119 talks about how BSF convert organic waste into biomass. These two ideas are essentially the same. I suggest the authors put them together rather than in separate paragraphs.

L148 Change to "...resulting DNA were evaluated..."

L338 There is a missing word or phrase between "research" and "did". Please check.

---

## Round 0.2 · accepted · Accept

In the revised version of manuscript all the suggested short comes have been incorporated by the authors, that were suggested by the experts of the field. The reviewer agreed to accept the manuscript after revision.

Reviewer 1 ·

Basic reporting

The authors have revised the manuscript according to my comments and suggestions.

Experimental design

no comment

Validity of the findings

no comment